# Smartphone-Based Maternal Education for the Complementary Feeding of Undernourished Children Under 3 Years of Age in Food-Secure Communities: Randomised Controlled Trial in Urmia, Iran

**DOI:** 10.3390/nu12020587

**Published:** 2020-02-24

**Authors:** Navisa Seyyedi, Bahlol Rahimi, Hamid Reza Farrokh Eslamlou, Hadi Lotfnezhad Afshar, Armin Spreco, Toomas Timpka

**Affiliations:** 1Student Research Committee, Urmia University of Medical Sciences, Urmia 571478334, Iran; ns.seyyedi@gmail.com; 2Department of Health Information Technology, School of Allied Medical Sciences, Urmia University of Medical Sciences, Urmia 571478334, Iran; hadi.afshar@gmail.com; 3Reproductive Health Research Centre, Urmia University of Medical Sciences, Urmia 571478334, Iran; hamidfarrokh@gmail.com; 4Department of Health, Medicine, and Caring Sciences, Linköping University, 581 83 Linköping, Sweden; armin.spreco@liu.se; 5Centre for Health Services Development, Region Östergötland, 581 85 Linköping, Sweden

**Keywords:** child malnutrition, nutritional education, mHealth intervention, randomised controlled trials, middle-income countries

## Abstract

The mothers’ nutritional literacy is an important determinant of child malnourishment. We assessed the effect of a smartphone-based maternal nutritional education programme for the complementary feeding of undernourished children under 3 years of age in a food-secure middle-income community. The study used a randomised controlled trial design with one intervention arm and one control arm (*n* = 110; 1:1 ratio) and was performed at one well-child clinic in Urmia, Iran. An educational smartphone application was delivered to the intervention group for a 6-month period while the control group received treatment-as-usual (TAU) with regular check-ups of the child’s development at the well-child centre and the provision of standard nutritional information. The primary outcome measure was change in the indicator of acute undernourishment (i.e., wasting) which is the weight-for-height z-score (WHZ). Children in the smartphone group showed greater wasting status improvement (WHZ +0.65 (95% Confidence Interval (CI) ± 0.16)) than children in the TAU group (WHZ +0.31 (95% CI ± 0.21); *p* = 0.011) and greater reduction (89.6% vs. 51.5%; *p* = 0.016) of wasting caseness (i.e., WHZ < −2; yes/no). We conclude that smartphone-based maternal nutritional education in complementary feeding is more effective than TAU for reducing undernourishment among children under 3 years of age in food-secure communities.

## 1. Introduction

Every country in the world is affected by one or more forms of child malnourishment, with undernourishment being more prevalent in low- and middle-income countries [1]. During the first three years of life, the period of 6–23 months of age is characterised by accelerated growth, predisposing the child to growth faltering [2]. Nutrient requirements are high in this age interval and cannot be met through breast milk alone. Meanwhile, children are not yet developmentally ready to consume the same meals as other members of the family. In this setting, complementary feeding refers to the timely introduction of developmentally appropriate, nutrient-dense foods in addition to breast milk. The World Health Assembly has committed to decreasing the global prevalence of wasting (measured with weight-for-height z score (WHZ) < −2) among preschool children to less than 5%, and the number of stunted (height-for-age z score (HAZ) < −2) children by 40% by the year 2025 [3]. Wasting is an indicator of acute malnutrition, i.e., the result of more recent food deprivation or illness [4,5], while stunting indicates long-term undernourishment. Stunting has been associated with negative physical and cognitive outcomes during early childhood as well as poor school performance, reduced economic productivity, and increased risk of non-communicable diseases into adulthood [6]. It has been estimated that the highest-burden is in low- and middle-income countries; 20% of children younger than 5 years fulfil the criteria for wasting [7], while globally there are 162 million stunted children in this age group [8]. 

Although the causes of undernourishment in children include various socio-economic and environmental factors [9], the mother’s nutrition-related knowledge, attitudes, and practices are among those of greatest importance [10]. Several studies have investigated prevention of child undernourishment through enhancement of mothers’ nutritional literacy [11,12]. Such interventions have comprised learning about exclusive breastfeeding, timely weaning, and complementary feeding with sufficient protein and energy density [13]. In food-secure communities, nutritional education has been found to improve linear growth among children aged 6–23 months, whereas the impact of interventions that provide food and nutritional supplements (with or without nutrition education) on linear growth has been found to be limited to food-insecure communities [14,15,16]. The lack of impact of education-only interventions in food-insecure communities has been explained by a lack of resources to implement recommendations in families with food insecurity. This implies that, regarding interventions addressing nutritional education, more knowledge is particularly needed about cost-effective interventions among mothers with undernourished children in food-secure communities [17]. Predictions have showed that approximately 80% of the world’s population will own a smartphone by 2020 [18]. The high prevalence of smartphone use provides novel possibilities to enhance health service delivery [19]. For instance, smartphone technology offers a cost-effective platform for distribution of evidence-based health information and behavioral change interventions [20,21]. Smartphones are found to be particularly suitable for implementation of patient-centred interventions and enhancement of patient self-management capabilities [22,23]. Smartphone interventions have been used to address breastfeeding practices [9,24] and obesity in preschool children [25,26] with promising results. However, few studies have used smartphone applications to target undernourishment in children [24] and, to our knowledge, to date no trials have evaluated nutritional education delivered by a smartphone application to support undernourished preschoolers. 

A pertinent research question is therefore whether smartphone technology can also be used to prevent child undernourishment by providing their mothers with nutritional education. The aim of this randomised controlled trial was to assess the effect of a smartphone-based maternal nutritional education program for the complementary feeding of undernourished children under 3 years of age in a food-secure middle-income community in Urmia, Iran. The hypothesis was that maternal nutritional education delivered by smartphones reduces acute child undernourishment. A recent study performed in the west Azerbaijan province in Iran reported a twofold higher prevalence of child undernourishment than the national mean (stunting 8.7%, wasting 7.5%, underweight 4.3% among children under 5 years of age). In Urmia (population 680.000; centre of the province), 5.7% of the children were stunted and 2.6% satisfied the criteria for wasting [27]. 

## 2. Materials and Methods

The study used a randomised controlled trial design with one intervention arm (smartphone application) and one control (treatment-as-usual (TAU)) arm (1:1 ratio). Participants were recruited between March 2018 and May 2018 in Urmia, Iran. The last follow-up of participants was performed in December 2018. The Urmia University of Medical Science Research Ethics Board approved the study design (Approval code: IR.UMSU.REC.1396.291). The study was registered in the Iranian Registry of Clinical Trials (reference IRCT20190705044103N1) at 2019-07-14.

The primary outcome measure was change in wasting status measured by the WHZ indicator. The secondary outcome measures were the change in wasting caseness (WHZ < −2; yes/no) and the change in the mothers’ nutritional literacy with regard to critical knowledge, feeding attitudes, and nutritional practice. The instrument used to measure nutritional literacy (Appendix A) was based on WHO recommendations and has been used in several previous studies [28,29,30,31,32]. It was translated from English to Persian using a back-translation procedure in which ambiguous expressions were corrected. The study also evaluated changes in underweight status (measured by the WAZ indicator), stunting status (the HAZ indicator), underweight caseness (WAZ < −2; yes/no), and stunting caseness (HAZ < −2; yes/no). 

All children seen at one well-child clinic in Urmia from April 2018 to May 2018 were assessed for study eligibility. The inclusion criteria for the study were children aged < 3 years with moderate or severe undernourishment with regard to wasting (WHZ < −2), and/or underweight (WAZ < −2), and/or stunting (HAZ < −2). The weight of the child was obtained using a calibrated electronic scale with a precision of 0.01 kg (10 g). Depending on the age and ability to stand, the child’s length (recumbent) or height (standing upright) was measured to the nearest 0.1 cm. Child–mother pairs were excluded if the child or mother was diagnosed with a medical disorder requiring treatment and if the mother did not possess a smartphone. Based on weight, height, and specific age data, WHZ, WAZ and HAZ were calculated using WHO standards and each child was classified with regard to undernourishment status. If a child–mother pair met the inclusion criteria and no exclusion criteria, a nurse forwarded the mother to a research assistant for information about the study, verbally and in writing. If the mother consented to study participation, her contact details were obtained and a consent form was signed. When participants entered the study, baseline socio-demographic data were collected on age and sex of the child and mothers’ age, education, employment status, location of residence, and household assets (Appendix A). In both study arms, child malnutrition indices (WHZ, WAZ and HAZ) and mothers’ nutritional literacy were measured using identical procedures before and after the 6-month intervention period. 

A smartphone application and educational content were developed based on the Maternity Guidelines for maternal and child health services issued by the Iranian Ministry of Health [33]. The application was to provide the mothers of newly-born children with an interactive healthy-child care guide structured into a set of learning topics. The topics included nutrition principles based on child age, behavioural methods for child feeding, child weaning time, introduction of complementary feeding, and mothers’ health. All learning materials included were cross-examined by expert nutritionists to ensure that they complied with the national guidelines on infant feeding in the first 3 years of life. A software engineer took 120 h to develop a preliminary version of the application. The user interface and functions of the smartphone application were refined using participatory design methods involving software developers, clinicians, medical informatics specialists, and end users in a process where functions were stepwise added and revised. Following repeated field tests, a final version of the application was implemented for the Android operative system for smartphones. The final application had two major features: (1) providing mothers with evidenced-based education on how to feed their child divided by the child’s age group, and (2) a chat function where clinicians answered the mothers’ questions regarding nutrition through the application.

In the TAU arm, mother and child pairs received routine health service treatment over a 6-month period. Regular monthly check-ups of the child’s development were provided at the well-child centre according to national recommendations (six check-ups during the study period) and standard nutritional information was forwarded to the mothers. The mother and child pairs were also visited by clinicians from the well-child centre for assessments to determine the age-related ability of the child. The mothers were asked checklist-based questions about their child’s play, learning, speech, behaviour, and motor abilities. In the intervention arm, the mothers had the educational application installed on their smartphones in addition to the services provided in the TAU arm. Each week messages were sent by email to the mothers reminding them to use the smartphone application. Answering the mother’s questions about nutrition and managing the reminder messages in the application for mothers, the intervention arm consumed on average about 25 min per week of clinicians’ time for each mother and child pair during the study period.

The required sample size was estimated at 96 mother–child pairs based on the following assumptions: α error (two sided)—0.05, 90% power, and 1:1 participant ratio between the intervention and TAU arms. We prognosticated the mean baseline WHZ score in the study population to −2.20 (0.50 SD) and wanted to detect a 15% post-intervention difference between the study arms. The required sample size to detect this difference was estimated to be 96 mother–child pairs. We recruited approximately 15% more pairs than the required sample size additionally (in total 110 pairs) to ensure that 96 pairs would complete the follow-up measures. The mothers, having consented to participate in the study, were allocated using Random Allocation Software 1.0 (https://random-allocation-software.software.informer.com/1.0/) to intervention or control groups in a 1:1 ratio. The randomisation was performed by a medical informatics researcher with no access to participant information and who did not participate in the enrolment process. The permuted block method of randomisation for a block size of four was used. Based on the limited size of the trial, this procedure was preferred to simple randomisation in order to maintain an adequate balance in the number of participants allocated to each of the study groups. Besides the investigators and data collectors who were blinded to group assignment, participants in the control group were blinded to the smartphone application provided to the intervention group

WHO Anthro software version 3.2.2 (www.who.int/childgrowth/software/en/index.html) was used to calculate the growth indicators (WHZ, WAZ, HAZ). We initially compared the baseline characteristics of participants in the intervention and control groups by applying chi-square tests and an independent t-test for mothers’ age. To compare the changes in the primary endpoint measure WHZ status in the smartphone and the TAU groups, independent t-tests were used. Regarding the secondary outcome measures, Chi-2 test was used to compare change in wasting caseness (WHZ < −2; yes/no), and independent t-tests to compare changes in the mothers’ nutritional literacy. Comparisons of changes in underweight status (measured by the WAZ indicator) and stunting status (the HAZ indicator) were made using independent t-tests, while between-group difference in underweight caseness (WAZ < −2; yes/no) change was analysed using the Chi-2 test, and in stunting caseness (HAZ < −2; yes/no) change by Fisher’s exact test. The level of statistical significance was set to *p* = 0.05. All statistical analyses were conducted with SPSS version 25. 

## 3. Results

### 3.1. Participant Flow

One-hundred of the 110 recruited mother and child pairs completed the study procedures: 50/55 pairs in the intervention and 50/55 pairs in the TAU group (Appendix B). Twelve percent of the participating mothers were employed outside the home and their mean age was 30.0 years (Table 1). The mean age of the undernourished children (49% boys and 51% girls) was 16.4 months (Table 2). Their distribution into age categories did not differ between the smartphone and control groups (*p* = 0.793). There were no statistically significant differences between the study groups regarding mothers’ sociodemographic characteristics or child age distributions.

### 3.2. Mothers’ Nutritional Literacy

Table 3 shows changes in the mothers’ nutritional literacy scores in the smartphone and TAU groups. The mothers in the smartphone group showed greater improvement compared with the mothers in the TAU group with regard to the nutritional literacy dimensions critical nutritional knowledge (*p* < 0.001), attitudes towards feeding (*p* = 0.001), and nutrition practice (*p* < 0.001). They also showed greater progress with regard to the total nutritional literacy score (*p* < 0.001).

### 3.3. Child Undernourishment

Table 4 shows changes in children’s undernourishment status indicators in the smartphone and TAU groups. The children in the smartphone group showed greater progress than the TAU group with regard to the primary endpoint measure wasting status (WHZ +0.34 (95% Confidence Interval (CI) ± 0.26); *p* = 0.011). The smartphone group also showed greater progress regarding underweight (WAZ +0.35 (95% CI ± 0.20); *p* = 0.001), and stunting (HAZ +0.34 (95% CI ± 0.21; *p* = 0.002) status.

Table 5 shows changes with regard to the children’s undernourishment caseness in the smartphone and TAU groups. The children in the smartphone group showed more recovery with regard to the secondary endpoint measure wasting caseness than children in the TAU group (*p* = 0.001). The children in the smartphone group also showed more recovery from underweight caseness (*p* = 0.002) and any caseness (*p* = 0.005), while no statistically significant change in caseness was observed for stunting.

## 4. Discussion

We observed positive intervention effects on wasting indicator status and wasting caseness among undernourished children under 3 years of age. Regarding stunting, a positive effect on the status indicator was recorded, but there was no effect on caseness. The latter finding was expected and can be explained by the relatively short intervention period and that effects on linear growth develop during longer periods of time. Regarding maternal nutritional literacy, effects were recorded for all studied aspects. We interpret these findings to suggest that a 6-month smartphone-based maternal nutritional education programme in complementary feeding is more effective than current routines for correcting undernourishment among children under 3 years of age in middle-income communities. The smartphone-based maternal education appeared to mainly contribute to the increase in nutritional knowledge. Of note, the remaining proportion of wasting caseness was 6% in the smartphone group, compared with 32% in the TAU group. In light of these findings, the poor development of nutritional literacy among the mothers in the TAU group is noteworthy. It may be questioned whether these mothers were able to use the materials on nutritional education forwarded to them from the well-child centre. Nutrient requirements are high among children in this age category (particularly between 6 and 23 months of age), and the timely introduction of appropriate nutrient-dense foods is essential for normal growth. It should be noted that in a recent meta-analysis [14], nutrition education-only interventions addressing complementary feeding were found to not affect wasting indicator status, while the impact on stunting status was small but significant. Explanations of the discrepancy between these findings and the results of the present study (clear effect on wasting and limited impact on stunting) include that the smartphone application combined nutritional information materials with a function for personal consultations with clinicians and that the present study population was restricted to undernourished children. Nonetheless, differences in intervention designs and study population definitions make it difficult to compare interventions addressing maternal nutritional education. Moreover, this randomised controlled trial suggests that acute undernourishment can be counteracted among small children by educating their mothers though smartphones about timely introduction of adequate nutrient-dense foods. However, this evidence is contingent on food availability, as the management of child undernourishment requires direct adjustments of food provision [15]. Furthermore, it should be remembered that smartphone ownership among mothers requires the achievement of a certain socioeconomic status, even in middle-income communities. If smartphone ownership is not uniformly distributed including to the poorest groups, the intervention may increase health disparity due to unevenly distributed effects which benefit those already better off [34].

Factors that are likely to have contributed to the intervention effect include the usability and usefulness of the smartphone application [35]. Inspired by Norman’s [36] usability design recommendations, the educational smartphone application was characterised by simplicity. Each mother was provided with task-oriented guidelines on complementary feeding in short text segments supported by images and a messaging function for personal consultations with clinical experts on child nutrition. For future long-term interventions on child undernourishment, social network functions, such as moderated discussion forums for mothers and facilities for formative self-assessments of health literacy, can be taken into consideration [37,38,39]. From the usefulness perspective, the educational application was characterised by an emphasis on content credibility and quality. For this purpose, cooperation with the national public health agency was established at an early stage, as parents are more likely to use educational materials if these are provided by a recognised health authority [40]. It must also be taken into consideration that even a well-planned smartphone application may convey inaccurate information to users [41,42]. A requirement for the application was, therefore, that that the nutritional guidelines were to be based on scientific evidence and WHO guidelines. This required the national public health agency to adjust their evidence-based guidelines for distribution through smartphones. Finally, the smartphone application was designed with the acknowledgement that not all undernourishment problems can be corrected by nutritional education, not even in food-secure communities. A recent Iranian study among mothers of children under 3 years of age showed more frequent detrimental feeding behaviours, such as emotional detachment and poor perception of the child’s communication signals, among mothers with an undernourished child than mothers with a normally developed child [43]. Furthermore, other studies [44,45,46,47] have highlighted the role of the mother–child interaction in the aetiology of some feeding disorders. Therefore, to widen the usefulness of the educational materials in the smartphone application, they were complemented with a function for personal consultations that allowed the mothers to seek advice for issues other than complementary feeding. 

This study has strengths and some potential limitations that need to be considered when interpreting the results. To our knowledge and based on a recent review [48], it is the first randomised controlled trial assessing the effect of smartphone-based maternal nutritional education addressing complementary feeding among undernourished young children of an age characterised by accelerated growth and growth faltering. Due to the low number of dropouts (<10%) and missing values (i.e., those mother–child pairs with some but not all outcomes at follow-up) completers-only analysis was found to be sufficient to assess the research question. Acute malnourishment (wasting indicator status) was used as the primary endpoint measure, while the additional outcome measures underweight and stunting have been confirmed to measure independently more long-term aspects of child development and growth [49]. Nonetheless, the short follow-up period implies that the long-term effects of the smartphone-based maternal education on undernourishment and linear growth need to be studied further. Moreover, the psychometric properties of the instrument used to measure the mothers’ nutritional literacy based on WHO recommendations have not been established, but the approach assessing knowledge, attitudes, and practices has shown acceptable face validity in multiple previous studies [28,29,30,31,32]. Finally, it should be taken into account that smartphone ownership requires some personal and socioeconomic resources, meaning that mothers burdened with severe personal issues or belonging to the poorest households in the community may not have been included in the study [34,50].

## 5. Conclusions

This study showed that smartphone-based maternal nutritional education in complementary feeding was more effective than TAU for reducing wasting status and caseness among undernourished children under 3 years of age in a food-secure community. Severely wasted children in a recent meta-analysis were reported to be 11 times more likely to die compared with normal children, and the mortality risk was elevated also for mild wasting [51]. The long-term effects of smartphone-based maternal nutritional education among young undernourished children need to be established in future studies.

## Figures and Tables

**Table 1 nutrients-12-00587-t001:** Characteristics of participating mothers (*n* (%)).

	Study Group ^$^		
	Smartphone	Control	
	*n* = 50	*n* = 50	*p* Value
**Formal education**			
**Elementary or high school**	30 (60)	38 (76)	*p* = 0.086
**Academic**	20 (40)	12 (24)	
**Employment status**			
**Stay-at-home parent**	43 (86)	45 (90)	*p* = 0.538
**Employed with salary**	7 (14)	5 (10)	
**Residency**			
**City**	16 (32)	24 (48)	*p* = 0.102
**Village or suburb**	34 (68)	26 (52)	
**Home ownership**			
**Yes**	15 (30)	13 (26)	*p* = 0.656
**No**	35 (70)	37 (74)	
**Home size**			
**3 rooms or fewer**	48 (96)	50 (100)	*p* = 0.495
**> 3 rooms**	2 (4)	0 (0)	
**Car ownership**			
**Yes**	21 (42)	17 (34)	*p* = 0.410
**No**	29 (58)	33 (66)	

^$^ The “Smartphone” group was provided a 6-month educational intervention through a smartphone application, while the “Control” group was provided treatment-as usual at the well-child centre.

**Table 2 nutrients-12-00587-t002:** Age of participating children (*n* (%)).

	Study Group		
Child Age (Months)	Smartphone	Control	Total
**0–6**	4 (8)	9 (18)	13 (13)
**7–12**	18 (36)	12 (24)	30 (30)
**13–24**	17 (34)	17 (34)	34 (34)
**25–36**	11 (22)	12 (24)	23 (23)
**Total**	50 (100)	50 (100)	100 (100)

**Table 3 nutrients-12-00587-t003:** Mothers’ nutritional literacy scores at baseline and score changes at follow-up in the smartphone and TAU groups (mean (± 95% Confidence Interval (95% CI)). *P*-values indicate difference in score change between the study groups.

	Mothers’ Nutritional Literacy Scores (Mean (95% CI))					
	Baseline Score		Score Change at Follow-Up			
	Smartphone Group *n* = 50	TAU Group *n* = 50	Smartphone Group *n* = 50	TAU Group *n* = 50	Between-Group Difference	*p*
**Critical knowledge**	19.94 ± 0.83	19.94 ± 0.71	5.56 ± 0.78	−0.94 ± 0.59	6.50 ± 0.98	<0.001
**Feeding attitudes**	68.78 ± 1.85	67.98 ± 2.23	7.52 ± 1.36	−1.86 ± 1.02	9.38 ± 1.70	<0.001
**Nutrition practice**	84.64 ± 2.77	82.40 ± 2.92	3.16 ± 1.02	1.04 ± 0.75	2.12 ± 1.27	0.001
**Nutritional literacy**	173.36 ± 3.83	170.32 ± 3.66	16.24 ± 1.97	−1.76 ± 1.58	18.00 ± 2.53	<0.001

**Table 4 nutrients-12-00587-t004:** Children’s undernourishment indicator status at baseline with regard to wasting (WHZ), underweight (WAZ), and stunting (HAZ), and status changes at follow-up in the smartphone and TAU groups (mean (95% Confidence Interval (95% CI)). *P*-values indicate difference in status change between the study groups.

	Children’s Undernourishment Indicator Status (Mean (95% CI))					
	Baseline Indicator Status		Indicator Status Change at Follow-Up			
	Smartphone Group *n* = 50	TAU Group *n* = 50	Smartphone Group *n* = 50	TAU Group *n* = 50	Between-Group Difference	*p*
**Wasting**	−1.82 ± 0.21	−1.80 ± 0.23	0.65 ± 0.16	0.31 ± 0.21	0.34 ± 0.26	0.011
**Underweight**	−1.89 ± 0.18	−1.81 ± 0.17	0.50 ± 0.15	0.15 ± 0.13	0.35 ± 0.20	0.001
**Stunting**	−1.48 ± 0.22	−1.37 ± 0.32	0.26 ± 0.15	−0.09 ± 0.14	0.34 ± 0.21	0.002

**Table 5 nutrients-12-00587-t005:** Children’s undernourishment caseness (z-score < -2) at baseline measured as wasting (WHZ), underweight (WAZ), and stunting (HAZ), and caseness change at follow-up in smartphone and TAU groups (*n* (percent)). *p*-values indicate difference in caseness change between the study groups.

Children’s Undernourishment Caseness (*n* (Percent))					
	Baseline Caseness		Caseness Change at Follow-Up		
	Smartphone Group *n* = 50	TAU Group *n* = 50	Smartphone Group *n* = 50	TAU Group *n* = 50	*p*
**Wasting**	29 (58)	33 (66)	26 (90)	17 (52)	0.001
**Underweight**	22 (44)	20 (40)	15 (68)	4 (20)	0.002
**Stunting**	17 (34)	15 (30)	6 (35)	1 (7)	0.088 *
**Any caseness**	50 (100)	50 (100)	33 (66)	19 (38)	0.005

* Fisher’s exact test was used.

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
