# Peer review of "Smartphone-Based Maternal Education for the Complementary Feeding of Undernourished Children Under 3 Years of Age in Food-Secure Communities: Randomised Controlled Trial in Urmia, Iran"

_nutrients, 2020, doi:10.3390/nu12020587_

Round 1

Reviewer 1 Report

I appreciate the opportunity to review this interesting and well carried-out clinical trial, and would like to provide the following comments to the Authors.

According to the description in the Methods section, there was a fair amount of work carried out to develop the simple to use and understand application, given the multi-disciplinary team involved. Likewise, there seems to be a fair amount of manpower involved to support it, as the availability of personal consultations to mothers was made available around the clock for the duration of the trial, involving clinicians that answered mothers’ questions. I think it will be useful for the authors to make reference to the costs involved, as it would be critical information if such a system were to be scaled up (as seems to be the intention).

An important issue related to the use of ehealth/mobile technologies is to report on the use of the service. I would like to suggest to the authors that they take a look at this point, and to evaluate if the intensity of use of the app / personal consultation chat was related to the outcome (including whether there was a minimum use required for the intervention to be successful, and perhaps exploring if there was a dose-response relationship between use of the app and main outcome.

The authors mention in their conclusion that the use of the educational smartphone application was highly accepted by the mothers. Was there any assessment of its acceptability by the health care providers?

I found it particularly troublesome to see that mothers who received TAU did not show any significant change in their Critical knowledge, Feeding attitudes, Nutritional practice, and Nutritional health literacy (as shown on Table 3). This should raise some serious concerns from the Health Care System, as this practice seems to be only wasting time of the health care providers and mothers receiving care, while doing nothing to improve clinical care and very little in relation to the children’s nutritional conditions (as shown on Table 4).

Reviewer 2 Report

General comments:

The authors have a great deal of data, but they did not make good use of it. The methods are not clearly described and the statistical tests, from the information provided, are not suitable. There is no clear hypothesis or research question. The outcome variables are not well stated. In some parts of the manuscript it seems the outcome variable is “mothers’ nutritional literacy”, in another parts it seems to be “change in the nutritional status of the children” from pre to post intervention. Why are there 2 correspondent authors? Is this the journal guidelines? The paper needs a full review by a native English reviewer. I am commenting only on the science and on the structure of the paper.

Details comments and suggestions:

Abstract

Page 1. Line 16. Specify the term “malnourishment” so it is clearer if the authors mean “under-nutrition” only or if the authors are using the term as “malnutrition” that covers both under-and over-nutrition situations. Page 1. Line 18. If “malnourished” means only “under-nutrition” outcomes such as stunting, wasting and/or underweight. Page 1. Line 22. Specify the WHO criteria? Are you looking at stunting (moderate= 2SD and severe+-3SD from the median for height-for-age?). Will you be looking at other undernutrition indicators as well? Page 1. Line 23. Does the intervention group receive TAU and the educational smartphone? What constitutes treatment-as-usual? The abstract needs clarification so it stands on its own. The authors need to clarify much earlier in the abstract that they are using stunting, wasting and underweight as their outcome variables. Usually, the way to report Weight-for-Height z-scores is WHZ (not WFH). Similarly, use HAZ (height-for-age z-scores) and WAZ (weight-forage z-scores)

Introduction

Page 1. Line 35. Do not use the word “malnourishment”. Use under-nutrition if focusing on stunting, wasting and underweight Use malnutrition if focusing on both under-and-over nutrition (stunting, wasting, underweight, overweight and obesity).

Unless the term is well defined, I am unable to pinpoint if the initial sentence of the introduction “Every country in the world is affected by one or more forms of child malnourishment” is correct or not. The introduction needs to be organized in a more coherent way. This is a good piece of research that is not well written which is a pity. I suggest the following structure for the introduction: Brief definition of stunting, wasting and underweight Health consequences of stunting, wasting, and underweight throughout the life course. Global situation on stunting, wasting and underweight What biocultural factors seem to impact on stunting, wasting and underweight? What seems to help? Then introduce the research on smartphones. Current status of stunting, wasting and underweight specifically in Iran.

Materials and Methods

Aims of this study. The authors state that: “The aim of the randomized controlled trial was to assess the effect of smartphone-based maternal 76 nutritional education directed at complementary feeding on malnourished children below 3 years…” but there is no research questions or hypothesis? Please state these.

What constitutes “treatment-as-usual”? The authors mention as a secondary outcome changes in stunting status. It is very unlikely that stunting status changes in 6-months and not possible to attribute any change to the intervention. Please comment on this, based on the literature regarding stunting. We need more information on the educational level of the mother and how the authors have controlled for that. How are changes in maternal nutritional health literacy measured?

Results

It is unclear if the outcome of this study is the changes in nutritional status of the children (stunting, wasting, and underweight) or changes in maternal nutritional health literacy. Please clarify. Table 3 shows changes in mothers’ nutritional health literacy, but we are not clear how the categories of “critical knowledge”, feeding attitudes” etc are assessed and quantified. It’s not clear what statistical tests the authors used to compare results between pre-and post-interventions, within each of the groups. Independent t-tests are not adequate for a repeated measures scenario.

Reviewer 3 Report

Thank you for your contribution to the journal. I have read the manuscript and have several questions and suggestions for improvement. First, I recommend organizing the description of the methodology, you should separate Participants and Study Design with Figure from Appendix A with the name of groups as in manuscript, Screening and Data Collection, Measures as anthropometrics, mother nutritional literacy (NL); statistical analysis. There is no clear description of NL and no explanation as to what you understand as NL. How often mothers from the TAU group visited centre? How this could affect the results?

Please explain/improve:

  • data about child undernourishment (line 127-130) should be moved up, e.g line 93., in my opinion, it has no relationship with aim or hypothesis
  • 860 pairs were enrolled, but only 110 met the inclusion criteria. Which exclusion criteria were not met most frequently?
  • title 3.1. Recruitment and participant flow- is not proper for the section; There is no section in study subject
  • there are no legends under the tables
  • line 293 and 306: March-December, and then April-May you started? Recommendation: write clearly when you started when you finished; intervention over a 6-month period, so small inaccuracy
  • line 318-320 -"baseline socio-318 demographic data were collected on age and sex of children, mothers' age, education and 319 occupation, household assets and land ownership" are different from those in Table 1
  • Discussion - it should be stressed that smartphone-based maternal education mainly contributed to the increase of nutritional knowledge (30% vs. nutritional practice 3,7%)

Author Response

Authors’ response: We thank the reviewer for the most useful comments that have helped us improve the quality of the article. The corrections are marked in yellow in the revised manuscript.

First, I recommend organizing the description of the methodology, you should separate Participants and Study Design with Figure from Appendix A with the name of groups as in manuscript, Screening and Data Collection, Measures as anthropometrics, mother nutritional literacy (NL); statistical analysis. There is no clear description of NL and no explanation as to what you understand as NL.

Authors’ response: Thank you for these pertinent comments. The methodology section has been completely revised according to the suggestions from you and the other two reviewers. NL was understood and intervened upon following the WHO guidelines (see revised Methods section).

How often mothers from the TAU group visited centre? How this could affect the results?

Authors’ response: The mothers in the TAU group paid monthly visits to the well-child centre (n=6 during the study period). This information is added to the manuscript.

Data about child undernourishment (line 127-130) should be moved up, e.g line 93., in my opinion, it has no relationship with aim or hypothesis

Authors’ response: We agree. The methodology section has been thoroughly revised to accommodate this comment.

860 pairs were enrolled, but only 110 met the inclusion criteria. Which exclusion criteria were not met most frequently? 
Authors’ response: Thank you. The information requested has been added to the participation flow-chart.

title 3.1. Recruitment and participant flow- is not proper for the section; There is no section in study subject
Authors’ response: The title has been revised.

there are no legends under the tables

Authors’ response: The legends are placed (as we understand the journal format) above the tables.

line 293 and 306: March-December, and then April-May you started? Recommendation: write clearly when you started when you finished; intervention over a 6-month period, so small inaccuracy

Authors’ response: Thank you. The information requested has been added to the manuscript.

line 318-320 -"baseline socio-318 demographic data were collected on age and sex of children, mothers' age, education and 319 occupation, household assets and land ownership" are different from those in Table 1

Authors’ response: The revision asked for has been made.

Discussion - it should be stressed that smartphone-based maternal education mainly contributed to the increase of nutritional knowledge (30% vs. nutritional practice 3,7%)

Authors’ response: We agree. The information asked for has been added to the discussion section.